# Increase of 4-Hydroxybenzoic, a Bioactive Phenolic Compound, after an Organic Intervention Diet

**DOI:** 10.3390/antiox8090340

**Published:** 2019-08-24

**Authors:** Sara Hurtado-Barroso, Paola Quifer-Rada, María Marhuenda-Muñoz, Jose Fernando Rinaldi de Alvarenga, Anna Tresserra-Rimbau, Rosa M. Lamuela-Raventós

**Affiliations:** 1Department of Nutrition, Food Sciences and Gastronomy, School of Pharmacy and Food Sciences, University of Barcelona, 08028 Barcelona, Spain; 2Consorcio CIBER, M.P. Fisiopatología de la Obesidad y Nutrición (CIBERObn), Instituto de Salud Carlos III (ISCIII), 28029 Madrid, Spain; 3INSA-UB, Nutrition and Food Safety Research Institute, University of Barcelona, 08028 Barcelona, Spain; 4Department of Endocrinology & Nutrition, CIBER of Diabetes and Associated Metabolic Diseases, Biomedical Research Institute Sant Pau, Hospital de la Santa Creu i Sant Pau, 08041 Barcelona, Spain; 5Unitat de Nutrició Humana, Hospital Universitari Sant Joan de Reus, Departament de Bioquímica i Biotecnologia, Institut d’Investigació Pere Virgili (IISPV), Universitat Rovira i Virgili, 43007 Reus, Spain

**Keywords:** healthy diet, phenolic acid, 4-HBA, crossover study, carotenes, microbiota metabolites, intervention, humans, metals

## Abstract

Consumption of organic products is increasing yearly due to perceived health-promoting qualities. Several studies have shown higher amounts of phytochemicals such as polyphenols and carotenoids in foods produced by this type of agriculture than in conventional foods, but whether this increase has an impact on humans still needs to be assessed. A randomized, controlled and crossover study was carried out in nineteen healthy subjects aged 18–40 years, who all followed an organic and conventional healthy diet, both for a 4-week period. Analysis of biological samples revealed a significant increase on the excretion of 4-hydroxybenzoic acid (4-HBA), a phenolic metabolite with biological activity, after the organic intervention. However, no changes were observed in the other variables analyzed.

## 1. Introduction

Organic food consumption has been increasing yearly over the last decade due to growing public awareness of its environmental benefits and alleged healthy properties [1,2]. The general belief that organic produce is healthier due to a lower use of chemical agents, such as pesticides, fertilizers and antibiotics, [3] is supported by studies reporting lower concentrations of pesticide residues in individuals consuming organic food [4,5,6,7,8]. Differences in nutritional composition associated with the cropping system have also been found, but more studies are needed to draw conclusions [9]. Factors known to influence the nutritional composition of food include crop variety, geographical location, climatic conditions, soil type, season and state of maturity from harvest to storage. Organic food seems to have higher amounts of bioactive compounds such as polyphenols and carotenoids than conventionally produced food [10,11,12,13,14,15,16]. When exposed to a stressful environment, plants activate defense mechanisms. Accordingly, a lack of synthetic protectors (pesticides, chemical fertilizers, etc.) induces organic crops to produce phytochemicals. Phenolic acids represent one third of the phenol group in a diet, but also many of them are produced from dietary polyphenols through microbiota metabolism. Approximately 90% of polyphenols are not absorbed in the small intestine reaching the colon, where they are transformed to other compounds such as the phenolic acids [17,18]. In addition, lower concentrations of cadmium have been observed in organic versus conventional cereals [19], as well as differences in the content of fatty acids and proteins [20,21,22,23]. Among foods of animal origin, total polyunsaturated fatty acid (PUFA) and n-3 PUFA concentrations are higher in organically rather than conventionally produced milk [21]. A similar profile has been observed in meat, although the evidence is weak [23].

The few studies to evaluate the effect of organic foods on human biochemical parameters and health have employed methodologies with some limitations and provide inconclusive results [9], so further intervention studies are needed to corroborate their possible beneficial effects. Consumers of organic produce are associated with having a higher quality diet, lower body mass index (BMI), greater physical activity [24,25] and a generally healthier and more holistic lifestyle [26,27]. Thus, the question is the following: Are the consumers of organic food healthier due to their lifestyle or also because their diet has a superior nutritional value?

The aim of this study was to evaluate the effect of an intervention with organic diet versus a conventional one on biological parameters, inorganic elements, bioactive compounds, and phenolic acids and carotenes in healthy subjects.

## 2. Materials and Methods 

### 2.1. Study Subjects

Twenty-one healthy volunteers aged 18–40 years were included in the intervention, 19 of whom completed the study and two dropped out alleging personal reasons. Participants had previous interest in healthy diets and organic food, and they were recruited from the Food and Nutrition Torribera Campus of the University of Barcelona and surroundings. Exclusion criteria were history of cancer, cardiovascular diseases, hypertension and dyslipidemia, chronic illness or homeostatic disorder, as well as toxic habits such as tobacco and other drugs and an excessive alcohol intake. 

After approval of the protocol by the Ethics Committee of Clinical Investigation of the University of Barcelona (Barcelona, Spain), the study was registered (ISRCTN29145931). Each participant signed an informed consent prior to the start, which was conducted according to the principles of the Declaration of Helsinki.

### 2.2. Study Design

An open, crossover, randomized and controlled study was carried out (Figure 1). Each volunteer consumed an organic diet (OD) and a conventional diet (CD), both for 4 weeks, and received dietary advice to support adherence. Organic products represented at least 80% of the OD and no organic foods were allowed in the CD. In both diets, subjects were encouraged to follow a healthy Mediterranean diet with a similar food pattern. Additionally, during the OD intervention participants were given weekly vouchers from Ecoveritas S.A., as well as products (oil, wine, snacks and canned vegetables) from other organic food companies to facilitate dietary compliance. At the end of each intervention, the absence of differential dietary patterns was checked. Interventions were separated by a washout period of two months. The study was run in the Department of Nutrition, Food Science and Gastronomy of the Food Science and Nutrition Torribera Campus of the University of Barcelona (Spain).

### 2.3. Assessment of Diet and Physical Activity

Before the study, adherence to the Mediterranean diet and physical activity were measured through a 14-item questionnaire [28] and the validated Spanish version of the Minnesota Leisure-Time Physical Activity Questionnaire [29], respectively. Also, at baseline, participants were asked about the frequency of organic food and beverage intake. After each intervention, a 137-item semi-quantitative food frequency questionnaire was filled in with the help of the study staff to assess nutrient and food intake [30].

### 2.4. Anthropometric and Clinical Data Measurements

Body weight was measured using an electronic scale and height with a stadiometer. The BMI was calculated from body weight and height. Waist and hip circumferences were measured with a measuring tape accurate to 0.1 cm. The waist-hip ratio (WHR) was calculated from these parameters.

Diastolic and systolic blood pressure (DBP and SBP) and heart rate were measured in fasting conditions with an OMRON M6 monitor in triplicate at each visit.

### 2.5. Sample Collection

Fasting blood was collected before and after each intervention. Blood samples were collected from the arm via venipuncture using tubes containing ethylenediaminetetraacetic acid (EDTA). After centrifugation of blood samples at 1902× *g* for 15 min at 4 °C, plasma was obtained. In addition, 24 h urine was collected at each visit. Plasma and urine were stored at −80 °C.

### 2.6. Laboratory Evaluations

Biochemical analyses were performed by an external accredited laboratory (mdb.lad Durán Bellido) as follows. C-reactive protein (CRP) was assayed by an immunoturbidimetry method. The lipid parameters (high density lipoprotein (HDL), low density lipoprotein (LDL) and total cholesterol and triglycerides) were tested by an enzymatic method. Urea and uric acid were measured by enzymatic and enzymatic/chromogen methods, respectively, and creatinine by the Jaffe method (as modified by Larsen) [31]. The concentration of total proteins was quantified by a Biuret reaction to the final point and amount of albumin by a bromocresol green method.

### 2.7. Analysis of Inorganic Elements in Plasma

Plasma samples were digested with nitric acid (HNO_3_) (Instra, J.T. Baker) in Teflon reactors. After incubation at 90 °C overnight, Milli-Q water was added to the reactors. An aliquot was transferred into assay tubes and stored at 4 °C for the chromatographic analyses. The inorganic compounds (Inorganic ventures, Christiansburg, VA, USA) used as standards were the following: Iron (Fe), arsenic (As), copper (Cu), cadmium (Cd), uranium (U), lead (Pb), zinc (Zn), calcium (Ca), magnesium (Mg), potassium (K) and sodium (Na). Fe, As, Cu, Cd, U, Pb and Zn were analyzed by ICP-MS (NexIon 350D. Perkin Elmer, Waltham, MA, USA) and Ca, Mg, K, P and Na, by ICP-OES (Optima8300. Perkin, Waltham, MA, USA). The analyses were performed in the facilities of the CCIT (Centres Científics i Tecnològics) of the University of Barcelona.

### 2.8. Extraction and Quantification of Phenolic Acids from Urine

Urinary phenolic compounds were extracted by solid phase extraction using a Waters Oasis HLB 96-well plate 30 µm (30 mg; Waters Oasis, Milford, MA, USA) [32]. Chromatographic analysis of phenolic compounds was performed by ultra-high performance liquid chromatography tandem mass spectrometry (UHPLC-MS/MS), using an API 3000 triple-quadrupole mass spectrometer (Sciex, Framingham, MA, USA). The separation was carried out with Milli-Q water and acetonitrile (Panreac Quimica S.A., Barcelona, Spain) with 0.025% formic acid in both solvents (Scharlau Chemie S.A., Barcelona, Spain), according to a method validated by our group [32]. A Waters BEH C18 column 1.7 µm (50 mm × 2.1 mm) and an Acquity UPLC BEH C18 VanGuard pre-column 1.7 µm (2.1 mm × 2.0 mm) were used.

The pool of standards was prepared in synthetic urine and included 3-(4-hydroxyphenyl) propionic acid (3,4-HPPA), 4-hydroxybenzoic acid (4-HBA), 3,4-dihydroxyphenylacetic acid (3,4-DHPAA), 3-hydroxyphenylacetic acid (3-HPAA), dihydrocaffeic acid (DHCA), hippuric acid, homovanillic acid, caffeic acid (CA), m-coumaric acid (m-Cou), p-coumaric (p-Cou) and gallic acid (GA) (Sigma-Aldrich, St. Louis, MO, USA) and 4-hydroxyhippuric acid (4-HH) (Bachem Americas Inc, Torrance, CA, USA). Ethylgallate (Extrasynthese, Genay, France) was the internal standard.

### 2.9. Extraction and Quantification of Carotenoids from Plasma

Carotenoids were extracted from plasma samples by liquid–liquid extraction [33]. Chromatographic analysis of carotenoids was performed by high performance liquid chromatography with ultraviolet diode-array detector (HPLC-UV-DAD), using an HP 1100 HPLC system (Hewlett-105 Packard, Waldbronn, Germany) containing a quaternary pump, coupled to a DAD G1315B. The separation was carried out with Milli-Q water, methanol and methyl-tert-butyl ether (Panreac Quimica S.A., Barcelona, Spain), according to a procedure previously validated by our group [33]. A Waters reversed-phase column YMC Carotenoid S-5 µm (250 mm × 4.6 mm) and a precolumn YMC Guard Cartridge Carotenoid S-5 µm (20 mm × 4.0 mm) were used.

The standards used were α-carotene, *β*-carotene, and all-E-lycopene (Sigma-Aldrich, St. Louis, MO, USA) and 5-*Z*-licopene (CaroteNature GmbH, Ostermundigen, Switzerland). These were pooled and prepared in synthetic human plasma (Sigma-Aldrich, St. Louis, MO, USA).

### 2.10. Statistical Analysis

Normality of distribution was assessed by a Shapiro-Wilk test. A non-parametric Wilcoxon signed-rank test was used for all statistical analysis due to the small sample size and the non-normality distribution. First, baseline measures were compared to corroborate similar pre-intervention conditions. As no significant differences between interventions at baseline were observed, the final analysis was performed with post-intervention measures (*n* = 19). Baseline values of variables were calculated from the mean of 38 observations (2 measurements for each subject). Differences were considered statistically significant when *p*  <  0.05. Statistical analysis was performed using SPSS Version 23.0 for Windows (SPSS Inc, Chicago, IL, USA).

## 3. Results

### 3.1. Participant Characteristics

Table 1 shows the baseline characteristics of participants. Nineteen healthy subjects (9 males and 10 females) completed the study. Approximately three out of every four individuals were occasional consumers of organic products (foods and beverages). The mean age was 30 years and subjects were physically active. The baseline adherence to the Mediterranean diet was high in 7 individuals (≥ 10 points); moderate in 11 (6–9 points) and low in 1 (≤ 5 points).

Baseline anthropometric (weight, BMI, waist and WHR), clinical (DBP, SBP and heart rate) and biochemical (CRP, HDL, LDL, total cholesterol, triglycerides, urea, creatinine, uric acid, total protein and albumin) measurements are also given in Table 1. The baseline concentrations of inorganic elements and bioactive compounds (phenolic acids and carotenes) are available as Appendix A, respectively.

### 3.2. Mean Dietary Composition of Participants During the Interventions

Participants followed a similar dietary pattern in both interventions (Table 2), although the OD was lower in protein (*p* = 0.036) and fish/seafood (*p* = 0.042). The mean proportion of macronutrients was the same in both diets (57% carbohydrates, 24% fats and 19% proteins). A borderline *p* was obtained comparing dairy products and vegetables (*p* = 0.051 and 0.055). However, the differences between both diets considering individual food were not significant (data not shown). In addition, a significantly lower amount of calcium and phosphorus was ingested in the OD.

### 3.3. Physiological Parameters of Participants After the Interventions

Table 3 shows anthropometric, clinical and biochemical data of the participants after following the OD and CD.

### 3.4. Inorganic Elements in Plasma

No significant differences were observed in the plasmatic concentration of minerals and heavy metals between the two diets (Table 4).

### 3.5. Phenolic Acids in Urine

Several polyphenols, mainly phenolic acids generated by microbiota metabolism and their derivatives, were evaluated in urine after the interventions (Table 5). A significant increase was observed in 4-HBA (*p* = 0.028) after the OD compared to the CD, but no changes were detected in the rest of the phenols.

### 3.6. Carotenoids in Plasma

No significant differences were observed in plasmatic concentrations of carotenes (Table 6).

## 4. Discussion

A randomized, controlled and crossover pilot study with nineteen healthy subjects was carried out to assess whether following an OD for 4 weeks changes health parameters and biomarkers compared to a CD.

In this study, the phenol 4-HBA increased approximately three times at the end of the OD compared to the CD (*p* = 0.028). 4-HBA can come from a diet, nevertheless, the intake of food rich in this phenol, such as berries, beer, etc., did not change significantly between both interventions (data not shown). However, this compound is produced from anthocyanins catabolism, as a metabolite of pelargonidin [34,35,36], and it can be formed by the colonic microbiota [36,37]. The metabolite 4-HBA has shown anticancer and neuroprotective effects [37,38,39,40]. Moreover, this compound is a precursor of the coenzyme Q10, showing cardioprotective properties [41,42].

No significant differences in the urinary concentration of the rest of the phenols were observed between the two diets, although vegetable intake was borderline lower in the OD. Stracke et al. carried out a study in which healthy men consumed 500 g of organic or conventional apples for four weeks. Twenty-four hours after the last intake, polyphenol concentrations in plasma and urine were not higher in the organic consumers [43].

Studies on the carotenoid content in organically grown fruits have provided inconclusive results [13,19,44]. In the present work, no effects of the OD on carotenoid levels were detected. In contrast to our results, a previous observational study reported significant differences in both carotenes and other fat-soluble micronutrients after consumption of organic food [45].

No changes in the concentration of inorganic elements in plasma were observed after either intervention. According to other authors, organic agriculture does not affect dietary copper [45,46] or zinc absorption [45]. However, a cohort from the NutriNet-Santé study presented a higher level of magnesium after following an OD, whereas no differences were found in iron absorption [45]. Higher magnesium, iron and phosphorus levels have been described in organic versus conventional plant-derived foods [47]. In contrast, concentrations of cadmium have been reported to be lower in organic food, due to the type of plant fertilizer used, but lower levels in consumers of organic produce were not observed [19,45,48]. In the present study, cadmium was not detected in plasma, nor was lead or uranium. Marchioni et al. showed that the content of cadmium and lead in coffee was influenced by temperature and mass, respectively, but not by the type of crop [49]. Although uranium is used more in conventional than in organic agriculture [50,51], a higher uranium content was not evidenced in conventional produce [52]. We found calcium and phosphate intake was lower in the OD, likely due to a lower consumption of dairy products. Although previous studies have described a higher concentration of phosphate in conventional foods due to crop fertilizers [53], here no differences were detected in the plasma levels between the two diets. Previous findings from the Environmental Defense Fund indicate that organic foods are as likely as conventional foods to contain heavy metals, because the organic standard is focused on pesticides and not these contaminants [54].

ODs are generally believed to be healthier and to provide more bioactive compounds. Some authors have observed a higher concentration of some phytochemicals in organic food, but without considering their bioavailability. In addition, when assessing the nutritional value of food, other influential factors need to be considered, including crop variety, maturity, soil and climate. On the other hand, consumers of organic products tend to be more concerned with health-related issues than the general population, which can bias the results of observational studies.

Organic foods are appreciated for the limited use of synthetic compounds (fertilizers, pesticides and antibiotics) in their production. Nevertheless, conventional crops are also regulated in this respect, and long-term studies are required to corroborate the effect of these compounds on health. To date, evidence suggesting that organic products are more nutritive or healthier is still lacking. Therefore, further carefully designed research is needed to evaluate the effect of an OD on bioactive compounds in biological fluids and health-related biomarkers.

The strongest point of the current study is its crossover design and the evaluation of a dietary pattern instead of only one or a few foods. Also, few such clinical assays have been conducted to date, with most studies being observational. Limitations of the work include a small sample size, the short duration of interventions and some differences in dietary patterns between the two interventions. However, this may be considered a pilot study to assess the short-term effects of organic food consumption. The increase of the phenolic compound arising from microbiota metabolism (4-HBA) in consumers following the OD need to be corroborated by further research with a higher number of subjects, which may shed light on a potential mechanism and possible health beneficial effects. In addition, a better control of factors as crop variety, maturity, soil and climate would provide more reliable results.

## 5. Conclusions

This intervention study for only one month found a significant difference in the concentration of a phenolic acid, the 4-HBA, after the OD. No changes were observed in the rest of the bioactive compounds analyzed nor in the other health-related biomarkers considered, neither in the results of minerals and heavy metals. The relation between the organic or conventional foods consumed and the concentration of bioactive compounds in the organism should be further researched. Longer studies and with larger sample sizes could reach significant values in other biochemical and heathy variables, demonstrating the health benefits of an OD.

## Figures and Tables

**Figure 1 antioxidants-08-00340-f001:**
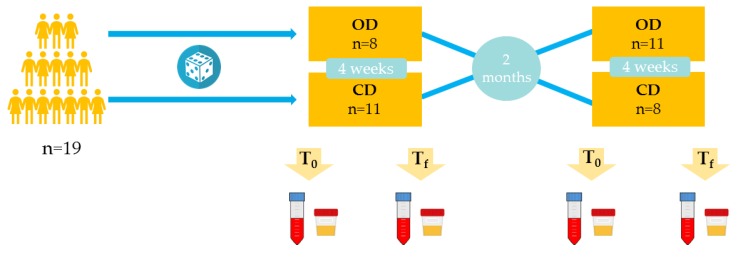
Study design. OD: Organic diet; CD: Conventional diet; T_0_: Initial time point (before interventions); T_f_: Final time point (after interventions).

**Table 1 antioxidants-08-00340-t001:** Baseline characteristics of the participants (*n* = 38).

Characteristics	
Males, *n* (%)	9 (47)
Occasional intake of organic products, *n* (%)	14 (74)
Age (years)	30 ± 1
Physical activity in leisure time (METS-min/week)	3814 ± 489
14-item MedDiet score (points)	9 ± 0.3
Weight (kg)	63 ± 2
BMI (kg/m^2^)	22.1 ± 0.4
Waist (cm)	76 ± 1
WHR	0.79 ± 0.01
DBP (mmHg)	75 ± 2
SBP (mmHg)	116 ± 2
Heart rate (bpm)	68 ± 2
CRP (mg/dL)	0.14 ± 0.03
HDL (mg/dL)	62 ± 3
LDL (mg/dL)	93 ± 5
Total cholesterol (mg/dL)	169 ± 6
Triglycerides (mg/dL)	69 ± 4
Urea (mg/dL)	29 ± 1
Creatinine (mg/dL)	0.80 ± 0.02
Uric acid (mg/dL)	4.60 ± 0.18
Total proteins (g/L)	72 ± 1
Albumin (g/L)	44 ± 0

Data are mean ± SEM unless otherwise specified. BMI: body mass index, WHR: waist–hip ratio, DBP: diastolic blood pressure, SBP: systolic blood pressure, CRP: C-reactive protein, HDL: high density lipoprotein, LDL: low density lipoprotein.

**Table 2 antioxidants-08-00340-t002:** Nutrient and food intake of participants in both diets (*n* = 19).

	OD	CD	*p*
Nutrient intake			
Energy (kcal/d)	1965 ± 203	2062 ± 204	0.070
Carbohydrates (g/d)	211 ± 21	220 ± 22	0.260
Total fat (g/d)	88 ± 9	92 ± 9	0.091
SFA (g/d)	22 ± 3	23 ± 3	0.064
MUFA (g/d)	45 ± 4	46 ± 4	0.136
PUFA (g/d)	12 ± 2	12 ± 1	0.136
Protein (g/d)	68 ± 9	72 ± 9	**0.036 ***
Ca (mg/d)	780 ± 111	847 ± 110	**0.024 ***
Mg (mg/d)	344 ± 37	353 ± 39	0.376
P (mg/d)	1352 ± 171	1433 ± 169	**0.018 ***
Fe (mg/d)	16 ± 1	16 ± 2	0.376
Food intake			
Dairy products (g/d)	192 ± 52	207 ± 50	0.051
Meat (g/d)	98 ± 20	102 ± 19	0.202
Eggs (g/d)	28 ± 3	31 ± 3	0.180
Fish and seafood (g/d)	56 ± 16	66 ± 16	**0.042 ***
Vegetables (g/d)	296 ± 32	366 ± 39	0.055
Fruits (g/d)	360 ± 68	377 ± 70	0.650
Nuts (g/d)	13 ± 5	12 ± 5	0.950
Legumes (g/d)	26 ± 5	26 ± 5	0.528
Cereals (g/d)	98 ± 11	98 ± 10	0.717
Oils (g/d)	40 ± 4	40 ± 4	0.317
Cocoa (g/d)	18 ± 5	21 ± 8	0.812
Coffee (g/d)	62 ± 16	59 ± 16	0.600
Tea (g/d)	22 ± 7	17 ± 7	0.106
Wine (g/d)	54 ± 23	62 ± 28	0.634

Data are mean ± SEM. **p*-value < 0.05. SFA: Saturated fatty acid, MUFA: monounsaturated fatty acid, PUFA: polyunsaturated fatty acid.

**Table 3 antioxidants-08-00340-t003:** Anthropometric, clinical and biochemical measurements after the interventions (*n* = 19).

	OD	CD	*p*
Anthropometric measurements			
Weight (kg)	64 ± 2	63 ± 2	0.365
BMI	22.1 ± 0.6	22.2 ± 0.6	0.352
Waist (cm)	76 ± 1	76 ± 1	0.549
WHR	0.80 ± 0.01	0.79 ± 0.01	0.822
Clinical measurements			
DBP (mmHg)	79 ± 2	73 ± 2	0.074
SBP (mmHg)	119 ± 4	118 ± 3	0.979
Heart rate (bpm)	70 ± 3	66 ± 2	0.326
Biochemical measurements			
CRP (mg/dL)	0.17 ± 0.07	0.26 ± 0.11	0.438
HDL (mg/dL)	62 ± 4	60 ± 4	0.301
LDL (mg/dL)	92 ± 9	90 ± 7	0.653
Total cholesterol (mg/dL)	168 ± 9	164 ± 7	0.494
Triglycerides (mg/dL)	66 ± 4	68 ± 4	0.421
Urea (mg/dL)	29 ± 2	29 ± 2	0.913
Creatinine (mg/dL)	0.80 ± 0.03	0.79 ± 0.02	0.763
Uric acid (mg/dL)	4.55 ± 0.29	4.68 ± 0.26	0.456
Total proteins (g/L)	73 ± 1	71 ± 1	0.145
Albumin (g/L)	44 ± 1	43 ± 1	0.136

Data are mean ± SEM. BMI: body mass index, WHR: waist–hip ratio, DBP: diastolic blood pressure, SBP: systolic blood pressure, CRP: C-reactive protein, HDL: high density lipoprotein, LDL: low density lipoprotein.

**Table 4 antioxidants-08-00340-t004:** Inorganic elements in plasma after the interventions (*n* = 19).

	OD	CD	*p*
Na (ppm)	2991 ± 20	2992 ± 19	0.445
K (ppm)	839 ± 14	844 ± 11	0.778
Ca (ppm)	88 ± 1	88 ± 1	0.717
Mg (ppm)	18 ± 0	18 ± 0	0.778
P (ppm)	104 ± 3	100 ± 3	0.136
Fe (ppb)	1252 ± 127	1339 ± 107	0.601
Zn (ppb)	778 ± 98	785 ± 46	0.376
Cu (ppb)	858 ± 64	856 ± 71	0.904
As (ppb)	4.35 ± 2.27	3.99 ± 0.95	0.221
Pb (ppb)	BLD	BLD	-
Cd (ppb)	BLD	BLD	-
U (ppb)	BLD	BLD	-

Data are mean ± SEM. BLD: Below limit of detection.

**Table 5 antioxidants-08-00340-t005:** Urinary phenolic acids excretion after the interventions (*n* = 19).

	OD	CD	*p*
Phenylacetic acids			
3,4-DHPAA (nmol)	90 ± 35	35 ± 9	0.42
3-HPAA (nmol)	943 ± 594	941 ± 440	0.717
Homovanillic (nmol)	154 ± 60	108 ± 27	0.868
Phenylpropionic acids			
3,4-HPPA (nmol)	10 ± 3	27 ± 12	0.407
DHCA (nmol)	1.2 ± 0.4	1.2 ± 0.4	0.955
Hydroxybenzoic and derivatives			
4-HBA (nmol)	205 ± 123	70 ± 35	**0.028 ***
4-HH (nmol)	471 ± 225	212 ± 85	0.306
Hippuric (nmol)	1281 ± 235	1463 ± 211	0.231
Hydroxycinnamic and derivatives			
CA (nmol)	7 ± 2	10 ± 2	0.349
m-Cou (nmol)	0.5 ± 0.3	0.26 ± 0.07	0.501
p-Cou (nmol)	0.3 ± 0.7	0.54 ± 0.19	0.554
GA (nmol)	0.48 ± 0.45	0.07 ± 0.03	0.878

Data are mean ± SEM. * *p*-value < 0.05. 3,4-DHPAA: 3,4-dihydroxyphenylacetic acid, 3-HPAA: 3-hydroxyphenylacetic acid, 3,4-HPPA: 3-(4-hydroxyphenyl) propionic acid, DHCA: dihydrocaffeic acid, 4-HBA: 4-hydroxybenzoic acid, 4-HH: 4-hydroxyhippuric, CA: caffeic acid, *m*-Cou: *m*-coumaric acid, *p*-Cou: *p*-coumaric acid, GA: gallic acid.

**Table 6 antioxidants-08-00340-t006:** Plasmatic carotenoids after the interventions (*n* = 19).

	OD	CD	*p*
*α*-carotene (nmol/mL)	0.39 ± 0.09	0.27 ± 0.06	0.552
*β*-carotene (nmol/mL)	1.03 ± 0.24	0.95 ± 0.22	0.744
E-lycopene (nmol/mL)	0.7 ± 0.17	0.78 ± 0.18	0.913
Z-lycopene (nmol/mL)	0.15 ± 0.04	0.20 ± 0.05	0.379

Data are mean ± SEM.

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
