# Peer review of "Increase of 4-Hydroxybenzoic, a Bioactive Phenolic Compound, after an Organic Intervention Diet"

_antioxidants, 2019, doi:10.3390/antiox8090340_

Round 1
Reviewer 1 Report
This paper presents work from a small randomized controlled trial with a crossover design so that participants served as their own controls. The aim was to assess differences in a variety of biological parameters before and after a 4-week diet intervention that consisted of organic food vs. the control condition of usual dietary intake.
This work represents and interesting and important research question and the authors should be commended for what must have been an enormous amount of work, but the analysis and interpretation have some methodological issues that need attention.
Major comments:
Need more clarity and/or justification for the statistical methods used. The text states that the final analysis contains post intervention results but there are no n sizes in the table, so the reader cannot be sure if the results represent n=19 or n=19 X 2 (using the 19 participants two times—once after they received the control diet and once after they received the intervention diet). Most importantly, the authors should justify why they did not use a repeated measures analysis for a crossover design.
https://newonlinecourses.science.psu.edu/stat509/node/123/
A more complete dietary analysis would be preferable, especially in light of the borderline differences seen in biological carotenoid parameters. Was this simply due to differences in carotenoid intake? At a minimum, the lack of a more detailed nutrient analysis should be noted as a limitation.
In a related topic, vegetable intake is borderline lower in the OD, but carotenoids are borderline higher. This should be addressed as a discussion topic.
Minor comments:
1. Source population: based on the description, I assume that this was a convenience sample, but please note where the volunteers were recruited and under what conditions so that the reader can assess potential bias in the results.
2. The units for blood lipids are different in table 1 vs. table 3. Consistent reporting of units throughout the paper is imperative.
3. L173: Suggest changing “poorer” to “lower”
Author Response
REVIEWER 1
Comments and Suggestions for Authors
This paper presents work from a small randomized controlled trial with a crossover design so that participants served as their own controls. The aim was to assess differences in a variety of biological parameters before and after a 4-week diet intervention that consisted of organic food vs. the control condition of usual dietary intake.
This work represents and interesting and important research question and the authors should be commended for what must have been an enormous amount of work, but the analysis and interpretation have some methodological issues that need attention.
Major comments:
C1- Need more clarity and/or justification for the statistical methods used. The text states that the final analysis contains post intervention results but there are no n sizes in the table, so the reader cannot be sure if the results represent n=19 or n=19 X 2 (using the 19 participants two times—once after they received the control diet and once after they received the intervention diet). Most importantly, the authors should justify why they did not use a repeated measures analysis for a crossover design.
R1- As the reviewer suggested, the sample size used in each analysis was added in tables and clarified in the section of the “Statistical analysis” (section 2.10, lines 158 and 159).
According to comment about the statistical test used, we choose a non-parametric Wilcoxon test instead of a repeated measures analysis due to two reasons: i) the non-normality of variables, and ii) the small sample size. This point is justified in the lines 154-156 of section 2.10.
C2- A more complete dietary analysis would be preferable, especially in light of the borderline differences seen in biological carotenoid parameters. Was this simply due to differences in carotenoid intake? At a minimum, the lack of a more detailed nutrient analysis should be noted as a limitation.
R2- As the reviewer pointed out, more details about diet have been added:
Lines 179-180: “However, the differences between both diets considering individual food were not significant (data not shown).”
Lines 205-206: “nevertheless, the intake of food rich in this phenol, such as berries, beer, etc. did not change significantly between both interventions (data not shown)”
C3- In a related topic, vegetable intake is borderline lower in the OD, but carotenoids are borderline higher. This should be addressed as a discussion topic.
R3- Because of the changes observed in the results after revision and correction of the database, this trend is not detected. However, significant changes were reach in the case of a phenolic compound, 4-hydroxybenzoic acid. Manuscript has been modified and this fact has been highlighted.
In addition, the topic suggested was commented in the discussion: “No significant differences in the urinary concentration of the rest of phenols were observed between the two diets, although vegetable intake was borderline lower in the OD.” (lines 211-212).
Minor comments:
Source population: based on the description, I assume that this was a convenience sample, but please note where the volunteers were recruited and under what conditions so that the reader can assess potential bias in the results.
As the reviewer pointed, details about the recruitment and conditions of subjects have been added in the section 2.1 (lines 66-68): “Participants had previous interest on healthy diets and organic food, and they were recruited from the Food and Nutrition Torribera Campus of the University of Barcelona and surroundings.”
The units for blood lipids are different in table 1 vs. table 3. Consistent reporting of units throughout the paper is imperative.
The units for blood lipids have been corrected in tables 1 and 3 and throughout the manuscript.
L173: Suggest changing “poorer” to “lower”
As the reviewer suggested, changing “poorer” to “lower” has been done.
Reviewer 2 Report
45-47: It is not clear which specific foods are referred to by the authors (provide examples for higher calcium foods). Please reword to make.
49-51: Could this be moved to lines 41 and 42?
Lines 59-61: The objective needs to be more clear.
Line 74: Based on the description, it is a "cross-over trial". It is not clear to me why authors have worded the design this way!
line 75: Should be "organic diet OR conventional diet" right? If the answer is no, please make the statement clear.
Line 162: How did the authors measure "The baseline adherence to the Mediterranean diet"? It seems that the high, moderate, and low have been measured through the questionnaire, however, this is not clear in the metholdology, or the results sections.
Table 2: what is the unit of PUFA and MUFA?
Lines 261-262: The statement is not clear. The value of P (table 6) for alpha and beta carotene is 0.07. Usually these marginal values are obtained in small-sized samples.
Author Response
REVIEWER 2
Comments and Suggestions for Authors
C1- 45-47: It is not clear which specific foods are referred to by the authors (provide examples for higher calcium foods). Please reword to make.
R1- These lines are referred to the systematic review and meta-analyses carried out by Barański et al., where the groups of fruits, vegetables and cereals were included and analyzed separately. However, they observed significant differences in concentrations of cadmium for cereals, but not for vegetables and/or fruits. This clarification has been added in the manuscript (line 49).
C2- 49-51: Could this be moved to lines 41 and 42?
R2- The sentence “Factors known to influence the nutritional composition of food include crop variety, geographical location, climatic conditions, soil type, season, and state of maturity from harvest to storage” has been moved to the lines 39-41.
C3- Lines 59-61: The objective needs to be clearer.
R3- As the reviewer suggested, the objective was clarified (lines 60-62).
C4- Line 74: Based on the description, it is a "cross-over trial". It is not clear to me why authors have worded the design this way!
R4- As the reviewer comment, the design of the study was crossover and this characteristic was described. In addition, the study was open (the researchers and volunteers knew the intervention), randomized (the order of interventions was randomized) and controlled (each volunteer was their own control due to the crossover design).
C5- line 75: Should be "organic diet OR conventional diet" right? If the answer is no, please make the statement clear.
R5- Because of the crossover design of the study, to avoid the effects from differences between subjects, each volunteer carried out both interventions (organic and conventional diets or conventional and organic diets, according to the randomization). Therefore, we consider correct to say an organic diet (OD) and a conventional diet (CD).
C6- Line 162: How did the authors measure "The baseline adherence to the Mediterranean diet"? It seems that the high, moderate, and low have been measured through the questionnaire, however, this is not clear in the methodology, or the results sections.
R6- The questionnaire used to measure the adherence to the Mediterranean diet was a 14-item questionnaire and the details about its are described in the reference 28 (lines 91-92). This interview was performed at the baseline of the study, but no after interventions. The results were provided in the section 3.1. (lines 167-168). The aim of this test was to examine the dietetic habits of volunteers.
C7- Table 2: what is the unit of PUFA and MUFA?
R7- The units of PUFA and MUFA (grams per day) have been added in Table 2.
C8- Lines 261-262: The statement is not clear. The value of P (table 6) for alpha and beta carotene is 0.07. Usually these marginal values are obtained in small-sized samples.
R8- This statement has been removed according to the corrected results of manuscript.
Round 2
Reviewer 1 Report
The responses to reviewer comments have adequately addressed concerns.